# Efficient characterization of electrically evoked responses for neural interfaces

**Nishal P. Shah** *
Stanford University

**Sasidhar Madugula**
Stanford University

**Pawel Hottowy**
AGH University of Science and Technology

**Alexander Sher**
University of California, Santa Cruz

**Alan Litke**
University of California, Santa Cruz

**Liam Paninski**
Columbia University

**E.J. Chichilnisky**
Stanford University

## Abstract

Future neural interfaces will read and write population neural activity with high spatial and temporal resolution, for diverse applications. For example, an artificial retina may restore vision to the blind by electrically stimulating retinal ganglion cells. Such devices must tune their function, based on stimulating and recording, to match the function of the circuit. However, existing methods for characterizing the neural interface scale poorly with the number of electrodes, limiting their practical applicability. This work tests the idea that using prior information from previous experiments and closed-loop measurements may greatly increase the efficiency of the neural interface. Large-scale, high-density electrical recording and stimulation in primate retina were used as a lab prototype for an artificial retina. Three key calibration steps were optimized: spike sorting in the presence of stimulation artifacts, response modeling, and adaptive stimulation. For spike sorting, exploiting the similarity of electrical artifact across electrodes and experiments substantially reduced the number of required measurements. For response modeling, a joint model that captures the inverse relationship between recorded spike amplitude and electrical stimulation threshold from previously recorded retinas resulted in greater consistency and efficiency. For adaptive stimulation, choosing which electrodes to stimulate based on probability estimates from previous measurements improved efficiency. Similar improvements resulted from using either non-adaptive stimulation with a joint model across cells, or adaptive stimulation with an independent model for each cell. Finally, image reconstruction revealed that these improvements may translate to improved performance of an artificial retina.

## 1 Introduction

Recent advances in large-scale electrical and optical recording have made it possible to record and stimulate neural circuits at unprecedented scale and resolution [Jun et al., 2017, Kipke et al., 2008, Kerr and Denk, 2008, Stosiek et al., 2003]. These advances suggest the possibility of using electronic devices to restore functions lost to disease, or to augment human capacities [Wilson et al., 1991, Schwartz, 2004]. One such application is an artificial retina, which can provide a treatment for incurable blindness by electrically stimulating retinal ganglion cells, the output neurons of the retina [Stingl et al., 2013, Humayun et al., 2012, Lorach et al., 2015]. A high-fidelity device must encode a visual scene by electrically stimulating retinal neurons in a way that produces accurate and useful visual perception (Figure 1A).

However, to achieve this goal, the device must control the precise, asynchronous patterns of activity transmitted by multiple ganglion cell types to the brain. This will require first identifying the location and type of many individual ganglion cells in the patient's retina, then characterizing their responses to electrical stimulation. Previously, it has been shown that the location and types of individual retinal ganglion cells can be identified using recorded activity [Richard et al., 2015]. However, efficient characterization of electrical responses remains unsolved.

For example, in *ex vivo* experiments with primate retina, intended as a lab prototype for an artificial retina, characterization of neural response to stimulation of each of 512 electrodes individually (to avoid nonlinear interactions) requires about an hour, and scales linearly with the number of electrodes. Thus, with the advent of arrays that can stimulate thousands of electrodes [Dragas et al., 2017] with multi-electrode current patterns [Fan et al., 2018], naive response calibration may be too time-consuming for the clinic.

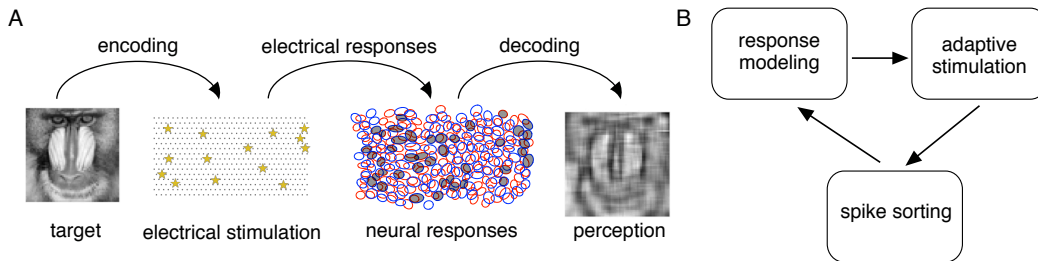

Figure 1: (A) Functional components of a retinal prosthesis. (B) Different steps in adaptive characterization of electrical response properties.

In the present work, new methods are proposed to efficiently characterize the electrical stimulation properties of a retinal interface, in a manner that may extend to other neural systems. Three novel steps are presented to calibrate the interface, based on the voltage recorded in response to electrical stimulation (Figure 1B):

- **Spike sorting in the presence of electrical artifact**: We develop a novel approach to estimate electrical artifacts, a key hurdle in spike sorting [Mena et al., 2017], in a subspace identified from past experimental data.
- **Response modeling**: We develop a model of electrical response properties of the target cells; this model incorporates as a prior the observed relationship between recorded spike amplitude and the threshold for electrical stimulation of a cell on a given electrode, from past experimental data.
- **Adaptive stimulation**: Inspired by previous work [Lewi et al., 2009, Shababo et al., 2013], we develop a method to exploit the data already recorded from a retina to optimize the choice of stimulation patterns for the next batch of measurements.

Finally, to make these methods most relevant for artificial vision, a modification is presented that minimizes error in the *reconstructed* visual stimulus, a more meaningful indicator of performance than the recorded spike counts.

Below, each algorithm is described separately in detail, with the results presented in aggregate after. Note that for simplicity, distinct notation is used for each algorithm; this notation is defined in the beginning of each section.

## 2   Spike sorting in the presence of stimulation artifacts

The goal of spike sorting is to identify spikes fired in response to electrical stimulation, and to distinguish spikes produced by different neurons. Spike sorting in the presence of electrical stimulation is difficult because the recorded spike voltages are corrupted by large stimulation artifacts with magnitude and duration comparable to those of the spike waveforms (see for [Mena et al., 2017] for examples). Hence, artifacts must be subtracted before identifying spikes. After artifact subtraction, spikes are identified based on waveforms previously recorded in the absence of electrical stimulation.

Previous work from [O'Shea and Shenoy, 2018] estimated the stimulation artifact by exploiting the artifact similarity for a given stimulation electrode, across different pulses, trials and recording

electrodes, but did not assign spikes to cells. [Mena et al., 2017] used previously recorded spike waveforms to jointly estimate the cellular activity and artifacts, assuming a smooth change in artifact with increasing currents. In comparison, this work identifies spikes by exploiting artifact similarity across stimulating electrodes and experiments, eliminating the need to track and separate spikes and artifact over a range of current levels, substantially reducing data requirements.

Let $\vec{y}_{a,r} \in \mathbb{R}^L$ be the $L$ dimensional recorded data on electrode $r$ when the stimulating electrode has amplitude $a$ ($L$ is the number of timesteps considered following the electrical stimulus). Using the artifacts estimated by applying the algorithm in [Mena et al., 2017] on previous experiments, an $n$ dimensional subspace $A_{a,d(r,e)} \in \mathbb{R}^{L \times n}$ is estimated for each stimulation amplitude ($a$) and distance $d(r,e)$ between the recording and stimulating electrodes. Hence the artifact is modeled as $A_{a,d(r,e)}\vec{b}_{a,r}$, with $\vec{b}_{a,r} \in \mathbb{R}^n$ estimated from the recorded data.

Let $\vec{x}_{c,a} \in \{0,1\}^L$ be the spiking activity of cell $c$ and $W_{c,r} \in \mathbb{R}^{L \times L}$ be the Toeplitz matrix consisting of shifted copies of a previously identified spike waveform on electrode $r$. Each neuron has at most one spike during the recording interval after stimulation, and the amplitude is exactly 1 when it spikes. This constraint is incorporated approximately by a softmax parameterization of $\vec{x}_{c,a}$ with an auxillary parameter $q_{c,a}$ allowing for the possibility of no spike and temperature $\tau$ determining the quality of approximation. Since neurons fire sparsely, an L1 norm penalty is applied on $x_{c,a}$ as well.

The artifact parameters $\vec{b}$ and spike assignments $\vec{x}$ are estimated by minimizing the penalized reconstruction error ($\mathcal{L}_{\text{spike-sort}}$) for a particular stimulating electrode $e$, the recorded voltage traces on multiple recording electrodes, and all the stimulating amplitudes simultaneously:

$$\mathcal{L}_{\text{spike-sort}} = \sum_a \sum_r \|\vec{y}_{a,r} - (A_{a,d(r,e)}\vec{b}_{a,r} + \sum_c W_{c,r}\vec{x}_{c,a})\|_2^2 + \lambda_{L1} \sum_c \|\vec{x}_{c,a}\|_1. \qquad (1)$$

For the results presented here, $L = 55$, $n = 9$, and cells with large amplitude on the stimulating electrode were used for spike sorting (roughly 10 cells per electrode). See Results and Appendix for details.

## 3 Response modeling

Given the spikes recorded in response to electrical stimulation, the goal of response modeling is to estimate the activation probability for each cell and electrode pair. The standard method, which involves estimating the response probability for each cell-electrode pair independently is presented first, followed by a joint model that incorporates priors from previous experiments.

For estimating these models, $N$ samples of electrical stimulus-response pairs $\{e_n, a_n, c_n\}_{n=1}^{n=N}$ are given, with stimulating electrode $e_n \in \{1, \cdots, N_e\}$, activated cell $c_n \in \{1, \cdots, N_c\}$, and current level $a_n \in \{1, \cdots, N_a\}$.

### 3.1 Independent model

This model assumes that there is no consistent relationship between recorded spikes and stimulation threshold across cell-electrode pairs. Thus, for each sample, the spiking probability is modeled as a Bernoulli distribution $P(R_n = 1) := \gamma_{e_n,a_n,c_n} = \frac{1}{1+e^{-(p_{e_n,c_n}(a_n-q_{e_n,c_n}))}}$, where $p_{e_n,c_n}, q_{e_n,c_n}$ are the parameters of the sigmoidal activation curve for the stimulating electrode $e_n$ and cell $c_n$. The parameters are inferred independently by using standard methods for maximizing the logistic log-likelihood for each cell electrode pair.

### 3.2 Joint model

Using prior data on the relationship between recording and stimulation from previously recorded retinas could lead to more efficient characterization of activation probabilities. Previous work [Madugula et al., 2017] suggests that for a given electrode, the recorded spike amplitude and the stimulation threshold are inversely related. The inverse relationship lies on different curves for axonal and somatic activation due to differences in channel density and geometry (Figure 3A). This section presents a model that jointly models this relationship across multiple cell-electrode pairs.

In the model, the activation threshold $q_{e,c}$ is related to the spike amplitude ($E_{e,c}$) using a reciprocal relationship, different for somatic or axonal activation ($T_{e,c}$) but common for all cell-electrode pairs

in a given retina (Equation 2). A Gaussian prior on the parameters of the reciprocal relationship $(x, y)$ is derived from previously recorded retinas (Equation 3, see Results):

$$q_{e,c} \sim \mathcal{N}(x_{T_{e,c}} + \frac{y_{T_{e,c}}}{E_{e,c}}, \ \nu^2) \tag{2}$$

$$\{x_T, y_T\} \sim \mathcal{N}(\mu_T, \Sigma_T); \quad T \in \{\text{soma}, \text{axon}\}. \tag{3}$$

Hence, the parameters of the model are given by

$$\Theta = \{\{p_{e,c}, q_{e,c}\}_{e=1,c=1}^{e=N_e,c=N_c}; \{x_j, y_j\}_{j \in \{\text{soma,axon}\}}, \nu\} \tag{4}$$

and the resulting model likelihood $(\mathcal{L}_{\text{model}})$ given by

$$\mathcal{L}_{\text{model}} = \Pi_n P(R_n | a_n; p_{e_n,c_n}, q_{e_n,c_n}) \Pi_{e,c} P(q_{e,c} | E_{e,c}; x_{T_{e,c}}, y_{T_{e,c}}, \nu_{T_{e,c}})$$
$$\Pi_{i \in \{soma,axon\}} P(x_i, y_i | \mu_i, \Sigma_i). \tag{5}$$

The goal is to estimate the posterior over the parameters $P(\Theta | \{R_n, e_n, a_n, c_n\}_{n=1}^{n=N})$. $\nu$ is learned but non-random, and other parameters are estimated by variational approximation of the posterior [Blei et al., 2017, Wainwright et al., 2008]. The posterior is approximated using a Gaussian mean-field variational distribution. This approximation is estimated by maximizing the evidence lower bound (ELBO) on the log-likelihood using the reparametrization trick [Kingma and Welling, 2013]. See Appendix for details.

## 4   Adaptive stimulation

In this section, the goal is to develop an algorithm that uses responses from prior stimulation within a retina to choose subsequent stimulation patterns, in closed loop. Since real-time closed loop experiments are generally not feasible using existing hardware, the experiment is assumed to run in multiple phases, with the algorithm choosing the entire collection of current patterns to stimulate in the next phase. The first phase is non-adaptive: each electrode and amplitude is stimulated $T$ times, for a total of $N_e N_a T$. In subsequent phases, parameter estimates from earlier phases are used to allocate a total of $N_e N_a T$ stimuli unevenly across electrodes and amplitudes.

The number of stimuli for each electrode and amplitude, $T_{e,a} \in \mathbb{Z}_+$, is computed by minimizing a loss function $\mathcal{L}$, that depends on the estimation accuracy of the stimulation probabilities. In this paper, the loss function is chosen as total variance in the estimate of response probability across electrodes and amplitudes $\mathcal{L} = \sum_{e,a,c} var(\gamma_{e,a,c})$, where $var(\gamma_{e,a,c})$ denotes the variance in estimate of activation probability. This condition is identical to A-optimality in optimal design literature [Atkinson et al., 2007], departing from the commonly used information-theoretic methods in neuroscience [Lewi et al., 2009, Paninski et al., 2007]; since the stimulation algorithms considered here choose only one electrode-amplitude combination at a time (Section 5), it is not necessary to account for the dependence of estimation error between different probabilities as in D-optimality.

For adaptively choosing the stimulations, the estimation error in response probabilities $\gamma_{e,a}$ after $T_{e,a}$ additional stimulations must be computed. Given the variance in estimate of the sigmoid parameters $(\theta_{e,c})$, the error in probabilities at individual current levels is given using Taylor expansion as $var(\gamma_{e,a,c}) \approx f_{e,a,c}^{'T} var(\theta_{e,c}) f_{e,a,c}'$, where $f'$ is the sigmoid derivative. If $T_{e,a}'$ are the number of previous stimulations, the variance of sigmoid parameters after $(T_{e,a} + T_{e,a}')$ stimulations is approximated using the inverse of the resulting Fisher information $I(\theta)^{-1}$. More concretely, the asymptotic variance of maximum likelihood estimate $\hat{\theta}$ is related to the inverse Fisher information computed using true parameter $\theta$. However as $\theta$ is unknown, the inverse Fisher information is approximated using the estimated parameters $I(\hat{\theta}_{e,c})$. The resulting optimization problem is:

$$\underset{T_{e,a}}{\text{minimize}} \quad \mathcal{L}_{\text{adapt-stim}} = \sum_{e,a,c} f_{e,a,c}^{'T} [I(\hat{\theta}_{e,c})]^{-1} f_{e,a,c}'$$
$$\text{subject to} \quad \sum_{e,a} T_{e,a} \leq N_e N_a T, \qquad T_{e,a} \geq 0 \ \ \forall e, a. \tag{6}$$

A soft-max representation of $T_{e,a}$ is used to minimize the unconstrained problem and the final approximate solution is quantized to give an exact (integer) solution (full details in Appendix).

# 5  Evaluation for artificial retina application

To optimize and evaluate these techniques in a manner that most accurately captures the function of an artificial retina, a metric for approximating the expected impact on visual function is developed.

Recent work [Shah et al., 2019] proposed that the perception from a collection of current patterns may be added linearly when they are combined by sequential stimulation at a rate faster than visual integration times. The stimulation sequence is chosen such that the response to each stimulation pattern is independent, and the final perception depends only on the total number of spikes generated in a temporal integration window. In this framework, the contribution of response probability estimation error from each cell-electrode pair to the accuracy of perception is estimated. Let $R_\gamma$ denote the observed spike response, given that the spiking probability is $\gamma$. When $\vec{d_c}$ denotes the linear decoding filter associated with activation of cell $c$, the cell's contribution to perception is given by $\vec{d_c} R_\gamma$. The expected change in perception when the cell response is generated by estimated probability $\hat{\gamma}$ instead of the true probability $\gamma$ is given by $\mathbb{E}_{\hat{\gamma}, R} \|\vec{d_c}(R_{\hat{\gamma}} - R_\gamma)\|^2 = \|\vec{d_c}\|^2 (var(R_{\hat{\gamma}}) + var(R_\gamma) + var(\hat{\gamma}))$, assuming $R_\gamma$ and $R_{\hat{\gamma}}$ are independent. The effect of inaccurate response probability estimation is mainly accounted for by the last term: $\|\vec{d_c}\|^2 var(\hat{\gamma})$.

In section 6.4, the performance for the retinal prosthesis application is evaluated using the above measure. The variance is either computed from the independent response model or the variational approximation of the joint response model. For adaptive stimulation, the optimization problem in Equation 6 is modified by weighing each term by the strength of the decoder $\sum_e \sum_a \sum_c \|d_c\|_2^2 var(\hat{\gamma}_{e,a,c})$.

# 6  Results

Extracellular recording and stimulation of primate retinal ganglion cells *ex vivo* using a 512-electrode technology [Litke et al., 2004, Frechette et al., 2005] were used to evaluate the performance of the algorithms. First, recorded voltages from 30 minutes of visual stimulation were spike sorted using custom software. The estimated spatio-temporal spike waveform for each cell was identified by averaging the recorded voltage waveforms over thousands of recorded spikes. Spike amplitude was measured on each electrode as the maximum negative voltage deviation, and spike shape was used to determine if the electrode was recording from the soma (biphasic waveform) or axon (triphasic waveform). Subsequently, electrical stimulation experiments were performed [Jepson et al., 2013] by passing brief ($\sim 0.1ms$), weak ($\sim 1\mu A$)) current pulses repeatedly through each electrode individually to identify the probability of eliciting a spike.

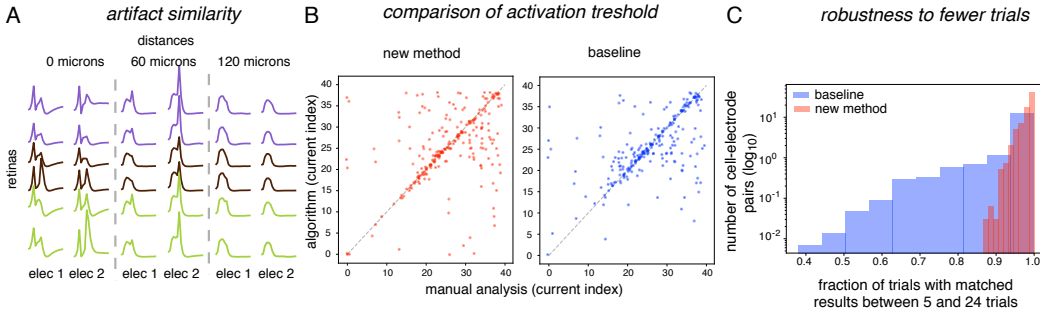

Figure 2: **Spike sorting** (A) Artifact recorded for a $0.68\mu A$ triphasic current pulse, on electrodes at different distances on a $60\mu m$ hexagonal grid. Lines in same column correspond to artifacts from different retinal preparations but the same electrode, lines with the same color indicate recordings from different pieces of retina from the same animal. (B) Comparison of estimated current index for 50% spiking probability for the algorithm (y-axis) and manual analysis (x-axis) for the new method (red) and the simplified method from Mena et al. [2017] (baseline, blue). Each of the 241 dots represents a cell-electrode pair. (C) Comparison of spike sorting results on 5 trials, when the 5 trials were analyzed independently, and as part of a total of 24 trials. Histogram across multiple cell-electrode pairs for the new method (red) and a simple form of a previous approach (blue) [Mena et al., 2017].

## 6.1 Spike sorting in the presence of stimulation artifacts

The performance of spike sorting was evaluated with voltage traces recorded in response to repeated electrical stimulation. For each of the recorded traces, stimulation artifacts were estimated by applying the simplified algorithm previously proposed in [Mena et al., 2017]. Briefly, the artifact estimate is initialized to the results obtained with a lower current amplitude, and then is updated by iterating between greedy spike estimation and artifact estimation. The estimated artifacts for different relative locations of stimulating and recording electrodes, across multiple experiments, are shown in Figure 2A. For a given stimulation current and a fixed distance between stimulating and recording electrodes, the artifact was similar across different stimulating electrodes, and across recordings.

Improved spike sorting was therefore implemented using the reduced space to regularize the artifact estimates. Performance was tested on the responses of one retina, with the artifact waveform basis estimated from a 9-dimensional approximation of data from 22 recordings (>99% of variance explained; see Appendix, Figure 6). Analysis of 24 repetitions for each electrode and amplitude revealed that the estimated activation threshold (current value that elicits a spike with probability 0.5) matched the value obtained with human analysis to within 67% in 161/ 241 of cell-electrode pairs (Figure 2B, left). In some cases the algorithm produced higher thresholds than the human, and vice-versa, but no large overall bias was observed. These results were comparable to the value of 173/241 of pairs obtained with the baseline method (Figure 2B, right). Thus, the new and baseline methods exhibited similar accuracy. To test whether the new method showed improved efficiency, both algorithms were applied in two cases: using 5 trials, or 24 trials, per stimulation pattern. The new approach resulted in greater consistency: spike times identified using fewer trials matched the spike times identified using more trials more closely than with the baseline method (Figure 6C) Thus, incorporating priors on artifact waveforms across recordings can allow for effective spike detection with limited data.

## 6.2 Response modeling

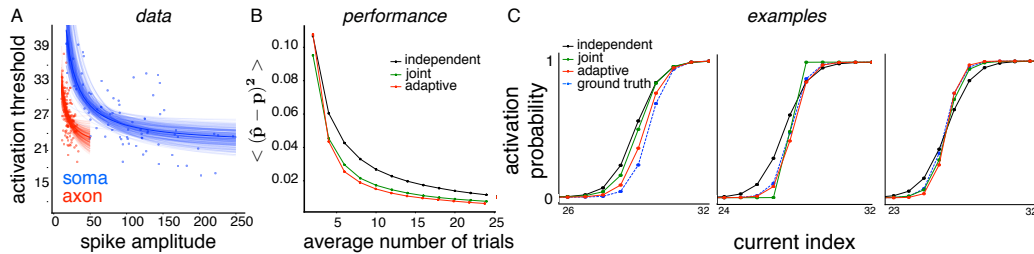

Figure 3: **Response models** (A) Relationship between observed spike amplitude (horizontal axis) and activation threshold (vertical axis) for electrodes recording from soma (blue) or axon (red). Each dot corresponds to a cell-electrode pair. Thin lines indicate fits obtained by different random samplings of a subset of cells (B) Squared error in probability estimates, averaged over multiple cell-electrode pairs (vertical axis) with different number of measurements using different methods. Measurements are done in batches, with a total of $2N_eN_a$ measurements per batch. (C) Estimated spike probability for a few cell-electrode pairs, showing improvement using the proposed methods.

To improve the efficiency of response model estimation, the effect of imposing priors on electrical properties of cells was examined. To test the impact of priors with ground truth and separately from spike sorting, neural responses were simulated based on activation probabilities estimated from a previously recorded retina. Two models were compared: (a) an *independent* model, in which spike probabilities are estimated without priors; (b) a *joint* model, in which spike probabilities are estimated with priors.

The key concept used for priors is that electrical stimulation thresholds for a cell on different electrodes should bear some relationship to the magnitude of the spike recorded on those electrodes, because both threshold and spike amplitude should depend inversely on the physical distance to the spike initiation region of the cell. Thus, for the joint model, the relationship between threshold and spike amplitude for both axonal and somatic activation was learned from a set of 199 cell-electrode pairs across 3 data sets (Figure 3A), using the spike sorting algorithm in [Mena et al., 2017]. The mean and variance of the parameters of the inverse relationship were identified by fitting curves with randomly sampled cell-electrode pairs.

The impact of using this prior was evaluated by measuring the responses from the simulated retina in batches, where an average of 2 measurements per electrode and amplitude were delivered in each batch. As the number of measurements increased, the joint model (green, Figure 3B, examples in Figure 3C and additional data set in Figure 7A) produced estimates of spike probability which were more accurate than estimates made by the independent model (black curve). Thus, a prior capturing the inverse relationship between spike amplitude and threshold can improve estimation of the response model.

## 6.3   Adaptive stimulation

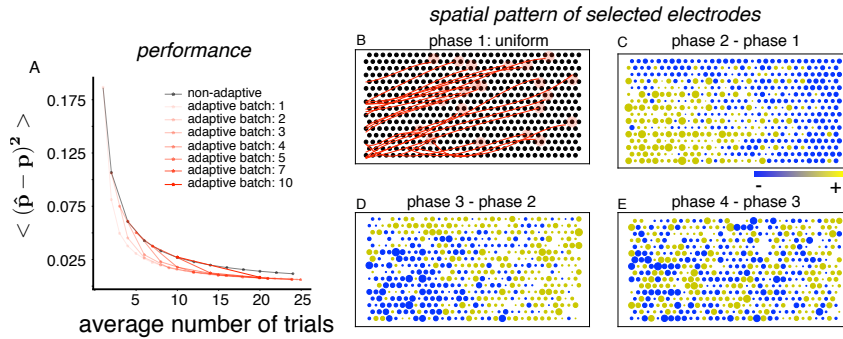

Figure 4: **Adaptive stimulation** (A) Adaptive stimulation with different batch sizes (red lines) and non-adaptive stimulation (black), both with the independent response model. Subsequent panels indicate spatial distribution of the electrodes selected by the adaptive algorithm. (B) First phase: All electrodes and amplitudes are selected uniformly (2 times each). Cell soma (red circle) and axon direction (red lines) are estimated from detected spike waveform across electrodes. (C) Second phase: The increase (yellow) and decrease (blue) in number of measurements in second (first adaptive) phase compared to the first (non-adaptive) phase. Size of circle indicates the magnitude of deviation. (D, E) Successive difference in the number of measurements for each electrode.

An alternative to regularizing based on previous retinas is to use feedback from measurements already made in the target retina to select the next electrical stimulus, and thus potentially improve estimation in closed loop. The effectiveness of this approach was tested using simulated data, to allow for more repetitions of electrical stimulation than were present in recorded data. An adaptive algorithm was developed using two stimuli on average per electrode and amplitude for each batch ($2N_eN_a$ measurements in total). After the first non-adaptive phase, the adaptive algorithm divides all the available capacity in the next batch across stimulations to minimize estimation error. With the simpler independent model, adaptive stimulation gave lower estimation error compared to the non-adaptive method (Figure 3B, red). The adaptive method with the independent model and the non-adaptive method with the joint model exhibited similar performance, suggesting that, with sufficient data, the computationally expensive adaptive stimulation can be replaced with better priors in non-adaptive stimulation. The adaptive method with the joint model performed similarly to the adaptive model with the independent model (not shown), possibly because the contribution of prior from previous experiments is reduced with better stimulus selection. The estimated response probabilities as a function of stimulation current for a few cell-electrode pairs after three phases of each approach are shown in Figure 3C, again showing similar performance of the two approaches.

In any adaptive method, a reduction in the number of adaptive phases reduces the computational burden, but could also reduce performance. To explore this tradeoff, the algorithm was tested with a smaller number of adaptive phases, and a corresponding increase in stimulation capacity per phase. A small number of adaptive phases typically yielded high estimation accuracy; for example, two phases with batch size 5 each yielded similar accuracy as 10 phases with batch size 1 each (Figure 4A).

Adaptive stimulation also revealed systematic spatial structure in the electrodes selected for stimulation. Compared to the uniform stimulation in the first (non-adaptive) phase, the electrodes on lower left side of the array were stimulated more frequently in the second (adaptive) phase (Figure 4C). This could potentially be explained by the geometry of the axons in the recording, which cross the array in a particular direction as they head toward the optic nerve (Figure 4B). Since cells could either be stimulated directly at the soma or indirectly at the axon, electrodes on the side of the array

with more axons would potentially stimulate more cells on average. Thus, these electrodes would contribute more to reducing error in estimation of response probabilities, and the adaptive algorithm preferentially selects them. However, in the third (adaptive) phase, the algorithm corrects itself and selects electrodes with fewer stimulated cells, presumably because estimation error remains high in those cells (Figure 4D). For subsequent phases, there is no obvious spatial structure in selected electrodes, presumably because the residual estimation is now similar across electrodes (Figure 4E). These observations were replicated in another retina (Figure 7B,C,D,E).

## 6.4  Performance for neural interface

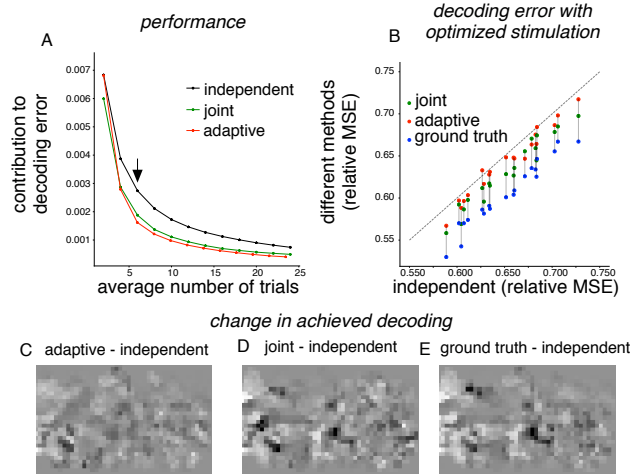

Figure 5: **Impact of electrical calibration on decoded stimulus** (A) Contribution of imperfect probability estimates to error in linear decoding (y-axis), measured by variance in estimated probabilities, weighted by decoder norm ($\frac{<\|d\|^2(p-\hat{p})^2>}{<\|d\|^2>}$) as a function of the average number of stimulation pulses (per electrode and amplitude, x-axis) for the three methods presented in the text. Independent and joint model same as Figure 3, but stimuli for adaptive method chosen using the modified objective. Black arrow indicates the trials at which the estimates are compared in (B). (B) Expected mean squared error of linearly decoded stimulus when probabilities from different methods are used for choosing the stimulation pattern. Each dot corresponds to a different white noise image. (C, D, E) The improvement in the expected linearly decoded stimulus using different probability estimates (over the independent model).

While the above techniques improve spike identification, response model estimation, and stimulus selection, a larger issue is how effective these improvements are for the function of the neural interface. In the case of vision restoration, performance ultimately depends on how each targeted cell contributes to vision, and on how well an actual image can be represented in the collection of cells.

To test functional impact in a way that accounts for how cells contribute to perception, the adaptive method was used with a modified error metric. Previous work [Shah et al., 2019] proposes that an artificial retina could linearly combine the expected perception from stimulation of different electrodes by temporally multiplexing within the integration time of the brain. Expected perception is inferred by assuming that the brain performs optimal linear reconstruction of the visual stimulus from retinal inputs. When electrode $e$ is stimulated at amplitude $a$, the change in perception due to error in the estimate of response probability is given by $\sum_c \|d_c\|^2 var(\hat{\gamma}_{e,ac})$, where $d_c$ is the optimal linear reconstruction filter for cell $c$. To evaluate adaptive estimation in this framework, the adaptive algorithm was modified to minimize the error in visual stimulus reconstruction, across electrodes and amplitudes (see Section 5). Applying the modified algorithm to simulated data revealed a faster decrease in the stimulus reconstruction error that is attributable to response probability estimation, compared to non-adaptive stimulation (Figure 5A). As before, the joint model with non-adaptive stimulation also outperformed the independent model.

To test functional impact in a way that captures variation in visual image structure, the spatial reconstruction of 20 different target images based on electrical stimulation was examined. Optimizing the electrical stimulation using estimated response probabilities for both the adaptive and joint

calibration algorithms (using the method in [Shah et al., 2019]) resulted in more accurate stimulus reconstruction compared to the independent model (Figure 5B,C,D). Thus, the gains from efficient characterization of electrical response properties are likely to translate into improved artificial vision.

# 7   Summary

This paper presents three novel methods to optimize the function of a neural interface, specifically, an artificial retina for treating blindness. Using large scale multi-electrode recordings from primate retina as a lab prototype, prior information and closed-loop approaches improved the accuracy and efficiency of spike sorting, response modeling, and stimulus selection. Notably, the computationally expensive closed-loop stimulation approach exhibited similar performance to a much simpler non-adaptive approach that uses prior information from previous experiments, highlighting the value of using large data sets to improve device function. Evaluation of image reconstruction revealed that these approaches improved overall function in terms of the quality of the visual image transmitted to the brain.

In principle, similar approaches may be useful in other neural systems (e.g. intra-Cortical micro-stimulation [Salzman et al., 1990] for proprioceptive feedback in motor prostheses [Salas et al., 2018]) and in other neural interfaces (e.g. optical recording and stimulation [Shababo et al., 2013]). With the advent of large-scale data sets, as well as the availability of motor and visual prosthesis technologies in many subjects, the methods developed here may be helpful in capturing similarities and differences among individuals and experiments.

# 8   Future work

Future improvements in electrical response calibration are possible by addressing current technical limitations and incorporating additional priors. For spike sorting, prior information about monotonic increase of activation probabilities with increasing currents could be incorporated. For wider applicability, the artifact estimation method should be extended to stimulation patterns that were not delivered in previous experiments. For response modeling, priors on the relationship between spike amplitude on multiple electrodes and activation threshold/slope differences between soma and axon may be useful [Jepson et al., 2013, Fan et al., 2018].

Several enhancements may be important in future work. First, for understanding the impact on artificial retina, analysis of linear image reconstruction [Warland et al., 1997, Stanley et al., 1999] could be extended to exploit more powerful nonlinear methods [Parthasarathy et al., 2017]. Second, it may be useful to evaluate response modeling, adaptive stimulation, and spike sorting together rather than independently. Finally, future work could focus on reducing computational cost in addition to the duration of electrical calibration.

## Footnotes

*Code: https://github.com/Chichilnisky-Lab/shah-neurips-2019

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

# 9 Appendix

## 9.1 Details on spike sorting in the presence of stimulation artifacts

Here, the spike sorting procedure presented in Section 2 is discussed in detail.

Let $\vec{y}_{a,r} \in \mathbb{R}^L$ be the recorded waveform of length $L$. The artifact is approximated in a subspace learned from previous experiments. For a given amplitude $a$, and relative distance to recording electrode $d(s,r)$, an $n$ dimensional basis is learned $A_{a,d(s,r)}$. The artifact is thus reconstructed using an $n$ dimensional learned parameter $\vec{b} \in \mathbb{R}^n$ as $A_{a,d(s,r)}\vec{b}_{a,r}$.

Let $\vec{x}_{c,a} \in \{0,1\}^L$ be the spiking activity and $W_{c,r} \in \mathbb{R}^{L \times L}$ be the matrix consisting of shifted copies of a previously identified spike waveform recorded on electrode $r$. The contribution of neural activity from cell $c$ to the recorded data is given by $W_{c,r}\vec{x}_{c,a}$.

Each cell has at most one spike during this recording interval, and when it spikes the amplitude is exactly 1. This constraint is incorporated by parameterizing $x_{c,a}$ as a softmax function of real valued $\vec{z}_{c,a}$ with temperature $\tau$ :

$$\vec{x}_{c,a,t} = \frac{e^{z_{c,a,t}/\tau}}{\sum_t e^{z_{c,a,t}/\tau} + e^{q_{c,a}/\tau}}$$

where $q_{c,a}$ is an auxiliary parameter. Since only a few cells are activated in response to electrical stimulation, a sparsity enforcing L1 norm penalty is applied on $\vec{x}$.

The artifact parameters $\vec{b}$ and spike assignments $\vec{x}$ are estimated by minimizing the penalized reconstruction error ($\mathcal{L}_{\text{spike-sort}}$) for a particular stimulating electrode $e$, the recorded voltage traces on multiple recording electrodes and all the stimulating amplitudes simultaneously:

$$\mathcal{L}_{\text{spike-sort}} = \sum_a \sum_r \|\vec{y}_{a,r} - (A_{a,d(r,e)}\vec{b}_{a,r} + \sum_c W_{c,r}\vec{x}_{c,a})\|_2^2 + \lambda_{L1} \sum_c \|\vec{x}_{c,a}\|_1$$

Optimization is performed using Adam (learning rate = 0.01), with $\lambda_{L1}$ chosen using cross-validation and temperature $\tau$ is reduced to 0.8 times its previous value every time the loss converges.

## 9.2 Details on the joint response model

Here, the joint response modeling procedure presented in Section 3.2 is discussed in detail.

### 9.2.1 Model

The responses are denoted by $R_n \in \{0,1\}$. Similar to the independent model,

$$P(R_n = 1) = \frac{1}{1 + e^{-(p_{e_n,c_n}(a_n - q_{e_n,c_n}))}} \tag{7}$$

where $p_{e_n,c_n}, q_{e_n,c_n}$ are the parameters of the sigmoidal activation curve for the stimulating electrode $e_n$ and cell $c_n$.

For each cell $c$ and electrode $e$, $E_{e,c} \in \mathbb{R}_+$ denotes the recorded spike amplitude and $T_{e,c}$ denotes whether the electrode $e$ is recording from the soma or axon, as determined by the spike shape. The spike threshold $q_{e,c}$ is modeled as a Gaussian distribution, with a separate relationship for soma and axons:

$$q_{e,c} \sim \mathcal{N}(x_{T_{e,c}} + \frac{y_{T_{e,c}}}{E_{e,c}}, \ \nu^2) \tag{8}$$

Further, the prior for $\{x,y\}$ is modeled with a two dimensional Gaussian

$$\{x_T, y_T\} \sim \mathcal{N}(\mu_T, \Sigma_T); \ \ T \in \{\text{soma}, \text{axon}\} \tag{9}$$

The parameters for prior distribution $(\mu_T, \Sigma_T)$ are estimated from stimulated electrode and cell pairs in previous experiments.

Hence, the parameters of the model for electrically evoked responses are given by $\Theta = \{\{p_{e,c}, q_{e,c}\}_{e=1,c=1}^{e=N_e,c=N_c}; \{x_j, y_j\}_{j \in \{\text{soma,axon}\}}, \nu\}$ and the resulting model likelihood ($\mathcal{L}_{\text{model}}$) is

$$\Pi_n P(R_n | a_n; p_{e_n,c_n}, q_{e_n,c_n}) \Pi_{e,c} P(q_{e,c} | E_{e,c}; x_{T_{e,c}}, y_{T_{e,c}}, \nu_{T_{e,c}}) \Pi_{i \in \{\text{soma,axon}\}} P(x_i, y_i | \mu_i, \Sigma_i).$$

### 9.2.2 Inference

The goal is to estimate the posterior distribution of model parameters given the recorded data $P(\Theta | \{R_n, e_n, a_n, c_n\}_{n=1}^{n=N})$. During inference, $\nu$ is treated as non-random and other parameters are estimated by variational approximation [Blei et al., 2017, Wainwright et al., 2008]. Let $z^{p_{e,c}}, z^{q_{e,c}}, z^{x_i}, z^{y_i}$ represent the variational parameters corresponding to parameters in $\Theta$. A mean-field variational approximation of the posterior is learned

$$P(\Theta | \{R_n, e_n, a_n, c_n\}_{n=1}^{n=N}) \approx \Pi_{e,c} q(z^{p_{e,c}}) q(z^{q_{e,c}}) \Pi_{i \in \{\text{soma,axon}\}} q(z^{x_i}) q(z^{y_i}). \tag{10}$$

The parameters of the variational distribution ($\phi$) are estimated by maximizing the evidence lower bound (ELBO) on the log-likelihood ($-\log \mathcal{L}_{\text{model}}$):

$$-\log \mathcal{L}_{\text{model}} \geq \mathbb{E}_{q(z)} \log P(R, Z) + H(q(z)). \tag{11}$$

As shown below, the first term of the joint probability corresponds to modeling electrically evoked spikes, the second term corresponds to modeling spike threshold from spike amplitudes, and the third term corresponds to the relationship between spike threshold and spike amplitude for all the cell-electrode pairs within a retina:

$$\mathbb{E}_{q(z)} \log P(R, Z) = \sum_{n=1}^{n=N} \mathbb{E}_{q_\phi(z^{p_{e_n,c_n}}), q_\phi(z^{q_{e_n,c_n}})} \log P(R_n, z^{p_{e_n,c_n}}, z^{q_{e_n,c_n}} | a_n)$$

$$+ \sum_{e,c} \mathbb{E}_{q_\phi(z^{q_{e,c}})} \log P(z^{q_{e,c}}, z^{x_{T_{e,c}}}, z^{y_{T_{e,c}}} | E_{e,c}, \nu_{T_{e,c}}) + \sum_i \mathbb{E}_{q_\phi(z^{x_i}), q_\phi(z^{y_i})} \log P(z^{x_i}, z^{y_i} | \mu_i, \sigma_i).$$

$$\tag{12}$$

The variational distributions are parameterized as Gaussians: $q_\phi(z^{p_{e,c}}) = \mathcal{N}(\phi_\mu^{p_{e,c}}, \phi_{\sigma^2}^{p_{e,c}})$; $q_\phi(z^{q_{e,c}}) = \mathcal{N}(\phi_\mu^{q_{e,c}}, \phi_{\sigma^2}^{q_{e,c}})$; $q_\phi(z^{x_i}) = \mathcal{N}(\phi_\mu^{x_i}, \phi_{\sigma^2}^{x_i})$ and $q_\phi(z^{y_i}) = \mathcal{N}(\phi_\mu^{y_i}, \phi_{\sigma^2}^{y_i})$, with $H(q)$ being the sum of Gaussian entropy corresponding to each variational parameter.

For maximizing the ELBO, the variational parameters are sampled using the re-parametrization trick [Kingma and Welling, 2013]: $z = \phi_\mu + \phi_{\sigma^2} \epsilon$, $\epsilon \sim \mathcal{N}(0, I)$. The empirical approximation of ELBO is computed by averaging 10 samples for $p, q$ and one sample for $x, y$. This approximation is maximized by stochastic gradient descent with norm clipping. At each step of the stochastic gradient descent, the objective function is evaluated over all samples, resulting in randomness only due to sampling of variational parameters. Finally, the posterior activation probabilties are estimated by averaging over 1000 random samples of $z^p, z^q$.

### 9.3 Details on adaptive stimulation

Here, adaptive stimulus selection procedure presented in Section 4 is described in detail.

The goal is to minimize the total uncertainty in activation probability estimates over all electrodes, amplitudes and cells ($\gamma_{e,a,c}$). The adaptive stimulation proceeds in batches, where a total of $N_e N_a T$ stimulations must be unevenly divided across electrodes and amplitudes. Let $T_{e,a} \in \mathbb{Z}_+$ denote the number of stimulations for electrode $e$ and amplitude $a$ in the next phase of the closed loop experiment. Hence, the optimization problem to be solved after each batch is given by :

$$\begin{aligned} \underset{T_{e,a}}{\text{minimize}} \quad & \mathcal{L}_{\text{adapt-stim}} = \sum_{e,a,c} var(\gamma_{e,a,c}) \\ \text{subject to} \quad & \sum_{e,a} T_{e,a} \leq N_e N_a T, \quad T_{e,a} \geq 0 \;\; \forall e, a. \end{aligned} \tag{13}$$

Define $X_a = [a, 1]$, $\theta_{e,c} = [p_{e,c}, -p_{e,c}q_{e,c}]$. Under this notation, the activation probability $\gamma_{e,a,c} = \frac{1}{1+exp(-\theta_{e,c}^T X_a)} = f(-\theta_{e,c}^T X_a)$. Let $T'_{e,a}$ denote the previous number of measurements for electrode $e$ and amplitude $a$. The Fisher information of $\theta_{e,c}$ after $(T_{e,a} + T'_{e,a})$ stimulations is computed using the chain rule as

$$I(\theta_{e,c}) = \sum_a (T_{e,a} + T'_{e,a})\gamma_{e,a,c}(1 - \gamma_{e,a,c})X_a X_a^T. \tag{14}$$

The asymptotic variance of the maximum likelihood estimate $\hat{\theta}$ is given by the inverse Fisher information at the true parameter values $I(\theta_{e,c})^{-1}$. Finally, the first-order expansion of $\gamma_{e,a,c}$ gives the variance of individual parameter estimates as $var(\gamma_{e,a,c}) \approx (f')^2 var(\theta_{e,c}) = Q_{e,a,c}^T var(\theta_{e,c})Q_{e,a,c}$, where $Q_{e,a,c} = \gamma_{e,a,c}(1 - \gamma_{e,a,c})X_a$. Normalizing the current levels $a$ between $-1$ and $1$ leads to better condition number for Fisher information and better approximation of the variance. After relaxing the integer constraint on $T_{e,a}$, the optimization problem is:

$$\underset{T_{e,a}}{\text{minimize}} \quad \mathcal{L}_{\text{adapt-stim}} = \sum_{e,a,c} Q'_{e,a,c}[\sum_{a'} (T_{e,a'} + T'_{e,a'})\gamma_{e,a',c}(1 - \gamma_{e,a',c})X_{a'}X'_{a'}]^{-1}Q_{e,a,c}$$

$$\text{subject to} \quad \sum_{e,a} T_{e,a} \leq N_e N_a T, \qquad T_{e,a} \geq 0 \ \ \forall e, a. \tag{15}$$

As the true parameter values are not unknown, the estimated probabilities in Equation 15 are used instead. These are estimated either by maximum likelihood on the independent model, or performing variational inference on the joint model. Re-parameterization of $T_{e,a}$ converts the constrained optimization problem into an unconstrained optimization problem. Specifically, a soft-max representation of $T_{e,a} = (N_e N_a T)\frac{e^{t_{e,a}/\tau}}{\sum_{e',a'} e^{t_{e',a'}/\tau} + e^{\delta_{e'}/\tau}}$ is used, where $\tau = 1000$ is the temperature parameter and $\delta_e$ is an auxillary parameter to allow for loose constraints. After minimizing the unconstrained problem using Adam optimization [Kingma and Ba, 2014], the exact integer solution for $T_{e,a}$ is obtained by rounding.

## 9.4 Approximation of artifacts in low dimensional space

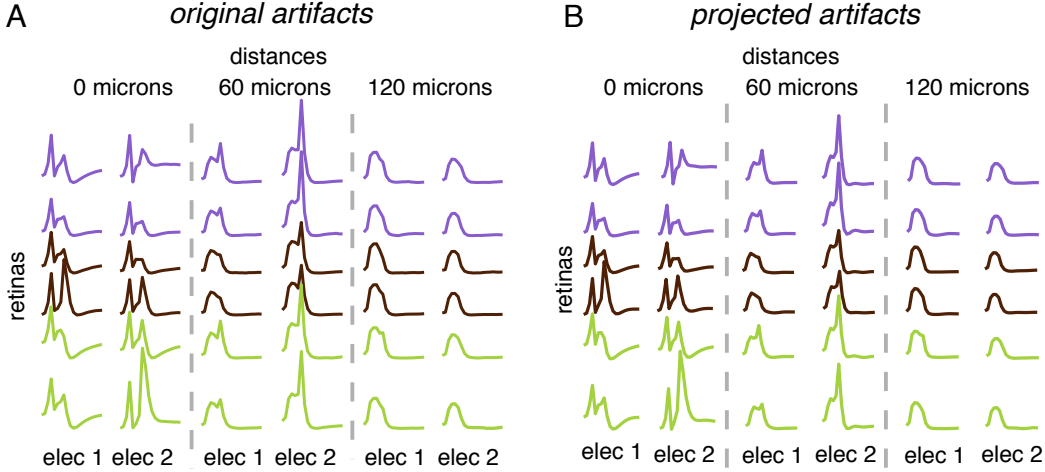

Figure 6: (A) Recorded artifacts (same as Figure 2A), (B) Reconstruction of artifacts from a 9 dimensional subspace. The projection retains most of the structure in the variation of artifacts.

## 9.5 Additional dataset

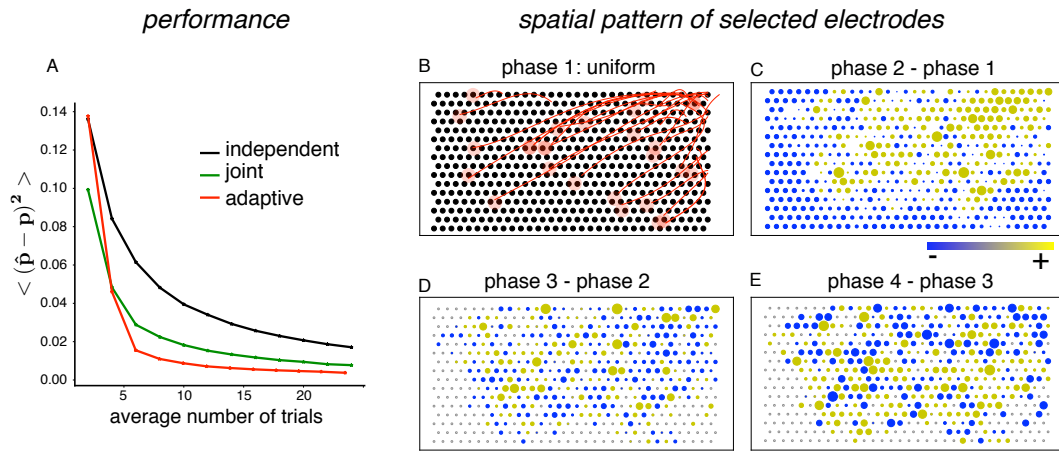

Figure 7: Results for another retina. (A) Conventions as in Figure 3B. (B-E) Conventions as in Figure 4(B-E).

