[Reviews · NeurIPS 2019]

Reviewer 1



Originality: The work is a novel combination of known techniques, which the authors clearly state in their introduction giving particular citation to Mena 2017 and Shah 2019. It is not entirely clear, based on the manuscript, that others have not attempted similar or otherwise effective methods to remove stimulus artifact. Perhaps this has not been done in retinal prostheses although this is not clear to this reviewer as those artifact reduction methods have been employed in a number of other electrophysiologic setups. Quality & Significance: Overall quality is high, authors fail to highlight many weaknesses in their approach. There are large-scale assumptions about how RGC create a perceived image which is more of a physiologic constraint and not critique-able in this setting. The most notable aspect of the work is that taking prior experimental data can provide as high a quality output as closed-loop approaches, which is perhaps the most important aspect of this work given the heavy shift in the literature to closed-loop devices. Clarity: The authors could provide code in addition to mathematical formulation, although this is a minor point.

Reviewer 2



Originality - The manuscript is lacking a prior relevant section in the introduction (although some prior methods are embedded within the text in the methods). Authors should include why “efficient characterization of responses remains unsolved” or why “complete probabilistic characterization of neural interfaces is infeasible.” What previous approaches have been used in the literature and how is their approach different? o Lines 33-35: “This will require first identifying the location and type of individual ganglion cells in the patient’s retina, then characterizing their responses to electrical stimulation.” Provide quantitative measures for readers to appreciate scale. o Lines 36-39: “However, efficient characterization of electrical responses remains unsolved. Neural responses to electrical stimulation are noisy and nonlinear, therefore in a device with thousands of electrodes, a complete probabilistic characterization of the neural interface is infeasible.” - Authors develop new models that leverage prior information to improve the calibration of the neural interfaces. In the introduction, authors should be more descriptive about the novelty of their work in contrast to previous approaches. - Spike sorting: o What is the novel approach based on and why does it solve the “key hurdle” in Mena et al., 2017 (line 46)? o From Section 2, the authors state: “The novel aspect of our method is to estimate the artifact in a subspace identified from previous experiments,” but it is not clear what distinguishes their work from Mena et al., 2017. For clarity, authors should first describe the approach in Mena et al., 2017 and how their approach improves/adds to this prior method. - Response modelling: o Authors develop a joint model to estimate the activation probability for each cell and electrode pair and use variational approximation to infer model parameters. o Line 49: “We use a model that incorporates as a prior the relationship between recorded spike amplitude and the threshold for electrical stimulation of a cell on a given electrode” Be more specific about the technical contribution, this is vague. o Line 87: Authors mention standard approach with the independent model. Is the joint model the authors’ proposed approach? If so, make it clear. o From the text, the results presented in Figures 3B-3C are obtained using the Mena et al. as the spike sorting algorithm (not their proposed sorting algorithm). Authors should make it clear in the figure caption, legends. - Adaptive Stimulation o Authors propose a data-driven sampling strategy to determine the stimulation electrode to stimulate within a predetermined budget. The sampling strategy is based on minimising the variance of the estimated model parameters. Quality - The technical content is generally sound, though some details for implementation are missing. Experiment protocols to validate proposed methods are generally well-designed to test the effects of proposed methods isolated from one another. Authors should come up with a nomenclature for the algorithms that specifies the specific models used in an algorithm based on: spike-sorting-response-model-stimulation, e.g. Mena, independent, non-adaptive (MIN). - Spike sorting o Results presented in Figure 2 (Mena-independent; Mena-joint) indicate while baseline and authors’ spike sorting algorithms perform comparably to manual analysis, the number of measurements is significantly reduced in the latter algorithm. o Are parameters initially estimated with the Mena et al. algorithm (line 66) used in the loss function for the signal reconstruction (line 78)? o Similar to lines 120-123, authors should state what each term in the loss function represents (note the information presented in line 76, \vec{y}_{a,r} = W_{c,r}\vec{x}_{c,a}). o How is the artifact filtered online based on the estimated parameters? o Figure captions, axis labels, legends should be more informative (Figure 2B what measure is being shown?) Include correlation coefficient between manual and algorithm; Figure 2C what is meant by “matching with fewer trials?” Also, y-axis is unlabelled. - Response modelling o Using the Mena spiking algorithm, improvements in response probability estimates are observed with the joint model that incorporates prior information when compared to an independent model. o Rationale for the joint model (Lines 219-222) should be mentioned in the relevant section for context (Section 3.2). - Adaptive stimulation improves performance of independent model to be comparable with the joint-nonadaptive model. - Neural interface o It is difficult to interpret the results in Figure 5 as the spike sorting, response model, and stimulation strategy are not clearly identified. Axis should be labelled accordingly to indicate what measure is being shown and figure captions more informative. What is the “hierarchical” algorithm and “modified objective” in Figure 5A? It is not clear what spikesorting algorithm and o What is being shown in Figure 5C-E? Also figures have no colour map. Clarity The paper is generally well-organised and readable but needs improvements in clarity with consistency in notation and nomenclature to properly identify based on the three components of the framework. A table with notations might also be useful. Line 70: What is c in W_{c,r} (line 70)? What is L Use more technical term than “dummy” parameter. Not clear what “responses” represented by R_n is (line 93), given that \vec{y}_{a,r} (line 70). It appears R_n is the same as \vec{x}_{c,a} (line 70)? Define space of E_{e,c} and T_{e,c} (lines 105-106) Section 3.1: Include equations for independent model. Consistency in notation: A few examples: q_{e,c} in line 94 and 107; x, y in lines 70 and 108; previously T denotes stimulation in axon or soma (line 106), but later T denotes the number of stimuli for electrode and amplitude (line 139). Lines 142-143: “computed by minimizing a loss function U, that that depends on the accuracy of parameter estimates.” What parameter estimates? Line 147: Revise “departing from the commonly used information-theoretic Lewi et al. [2009], …” Line 215-216: How does simulation translate to empirical data? “We simulate neural responses based on activation probabilities estimated by applying the existing sorting algorithm on a previously recorded retina.” Figure 3A Indicate soma/axon in legend in figure. Larger figure and makers to distinguish when in black/white. Also, for clarity labels “non-adaptive independent, non-adaptive joint, adaptive independent” to emphasise improvements from independent model in Figures 3A and 3B. Also, show results for adaptive joint, even if performance is similar – trade-offs between using either method. Significance Paper provides a useful contribution to the field of neural interfaces as it addresses the challenging problem of characterising the high dimensional neural response properties for more high fidelity device. Results from modification each component of the framework demonstrate improvements with authors' proposed methods. However, it is difficult to assess improvements to neural interface outcome as details are missing. Post-rebuttal: Read and appreciated authors' responses to main critiques from this reviewer and others, which should be included in a revised manuscript. Satisfied with most of the clarifications provided on prior work (particularly Mena et al.) and contributions of this work. Revising score upward.

Reviewer 3



The manuscript proposes new methods for spike sorting, response modelling and adaptive stimulation. The spike sorting algorithm minimizes the penalized reconstruction error by subtracting linear combinations of known artifacts and spike waveforms from the recorded signal. Response modelling is performed in two ways. Firstly, by modelling spiking probabilities as Bernoulli distributions and inferring its parameters by maximization of logistic log-likelihood, and secondly, by modelling spike amplitude and stimulation thresholds jointly across multiple cell-electrodes pairs. Adaptive stimulation is implemented as closed-loop algorithm that uses prior stimulation responses to choose subsequent stimulation patterns. The methods are based/inspired by the literatures and original. The reviewer did not check all equations in detail, however, used methods are valid and the reasoning of the authors is sound. The level of detail is ok. Authors evaluate the algorithms by performing one single ex vivo stimulation experiment of primate retinal ganglion cells. The preliminary results are promising; however, the significance of the results is still somewhat limited. The method, if transferable to other experiments, has potential and will be useful to other researcher. Please find additional comments below: #0: The title says “efficient” and “neural interfaces”. The manuscript does, however, not quantify efficiency. Also, the method is tested for retinal implants only. #93: Word repetition “the the” #143: Word repetition “that that” #190: How was the shape/duration of the stimulation? #205: Activation thresholds identified by the method are compared human results. What precisely do authors mean by “match the value”? #209: Authors again use the word “matched”. Please quantify. #246: What is the computational complexity of the algorithm? Pros and cons of the method are not discussed in enough detail. Specifically, in the context to large-data sets. Would recording hardware or other factors impact on the performance of the method? ======== Response to authors’ feedback: Thank you for the clarifications and for the willingness to characterize the pros and cons of the proposed method. The reviewer still thinks that the manuscript is above the threshold. But the question now is how the authors manage to present this variety of information in a short and clear way.

[Author Response · NeurIPS 2019]

Below are the responses to major issues raised by the reviewers; other issues (figures, equations, nomenclature, implementation and run-time details, code release) will be addressed in revision:

**Primary contribution of this paper (reviewer 2 and 3)** The reviewer comments were helpful here. We think the most important and unique contribution of this work is to leverage the distribution of model parameters from previous experiments for efficient characterization of the neural interface. Importantly, we focus on a unique large-scale interface operating at single-cell resolution. Our work unifies and extends previous studies on closed loop experiments [Paninski et al., 2007] and exploration of model architectures [Real et al., 2017] by incorporating a prior on model parameters from previous experiments in many animals, recorded over many years. As large-scale and high-resolution devices become more common, similar multi-animal datasets will likely be available. However, we are not aware of other work that matches the present work in resolution or scale of data. We would emphasize these points in revision.

**Application to other systems (reviewers 1 and 3)** Although the results are presented in the context of primate retina, the methods do not rely on specifics of the retinal circuitry, and we expect they would be useful in other neural systems as well. Specifically, the similarity of artifact shape across experiments is likely governed by impedance at the tissue interface. Also, the relationship between spike amplitude and stimulation threshold may be general, and may depend only on the spatial configuration of the electrode and the cell. We think that these methods are relevant to Intra-Cortical Micro-Stimulation (ICMS) [Salzman et al., 1990] for proprioceptive feedback in somatosensory cortex for motor prostheses [Salas et al., 2018], Optogenetics [Shababo et al., 2013] or as reviewer 1 suggests, for DBS (though current devices do not approach cellular resolution).

**Problem statement (reviewer 2)** In the context of a bi-directional retinal prosthesis, this work addresses one of the major outstanding problems regarding characterization of electrical response properties using a small number of measurements of electrical stimulation (i.e., efficiency - reviewer 3). In our lab prototype, identification of location and type for around 500 cells requires a few minutes of spontaneous activity recordings, performed in parallel across 512 electrodes. However, for electrical stimulation, each electrode needs to be stimulated in isolation to avoid nonlinear interactions ($\sim 1.5$ hours for 512 electrodes); this measurement thus scales linearly with the number of electrodes. The problem of electrical response calibration has not been addressed previously, primarily due to the much fewer number of stimulating electrodes in most existing devices (reviewer 2). However, with advent of larger arrays that can stimulate 1000s of electrodes [Dragas et al., 2017] with multi-electrode current patterns [Fan et al., 2018], naive response calibration in the clinic may be far too time-consuming. To make these devices usable, it will be necessary to substantially reduce the calibration time, making methods such as the ones presented here crucial.

**Relationship to prior work on neural interfaces (reviewer 2)** Even though we mention most of the prior works in the context of the methods, we failed to include enough information about prior works on spike sorting. Mena et al., 2017 use previously recorded spike waveforms to jointly estimate the cellular activity and artifacts, with multiple trials of a single stimulation current. A Gaussian process prior is used for smoothly extrapolating the artifact across current values. O'Shea et al., 2018 only estimates the stimulation artifact (they do not assign spikes to cells), exploiting artifact similarity for a given stimulation electrode, across different pulses, trials and different recording electrodes. In contrast, this work performs joint estimation of artifact and spikes, exploits the similarity of the artifact across experiments and stimulating electrodes, and does not require an increasing sequence of current values.

**Shortcomings (reviewer 1 and 3)** We agree that the limitation of the study should be addressed directly in the manuscript. (1) We focus on minimizing the number of electrical stimulations, it will be important in future to minimize computational runtime as well. (2) For spike sorting, linear super-position of spikes and artifacts fails when recording amplifiers saturate. (3) The artifact is characterized using stimulation of a given current pattern in previous experiments, thus, the method is inapplicable to novel stimulation patterns. (4) For response modeling, the relationship between single electrode spike amplitude and stimulation threshold must be generalized to use spike amplitudes and simultaneous stimulation from multiple electrodes. (5) The method should account for differences in activation curve slopes for axons and somatic activation. (6) Analysis of linear decoding [Brackbill et al., 2018, Warland et al., 1997, Stanley et al., 1999] for estimation of prosthesis performance should be modified to

Figure 1: **Another dataset (reviewer 3).** A, B Same as Figure 3B, 4C in paper.

incorporate nonlinear methods, which can yield higher performance [Parthasarthy et al., 2017]. (7) Response modeling and adaptive stimulation are validated only in simulation, where ground truth is available. (8) Each algorithm is analyzed in isolation, but the combined improvement from using all three should be evaluated (this is difficult due to lack of ground truth activation probabilities). These caveats will be included in the paper, subject to space limitations.

[Meta-Review · NeurIPS 2019]

In a conference that is all too often only ML and not enough neuro, it is refreshing to see a reasonably well-received neuro paper. This paper, which introduces new methods for spike sorting, response modeling and adaptive stimulation narrowly made the cut-off and will appear at this NeurIPS. There is not a whole lot of criticism to share beyond what the reviewers have already provided; this is a good paper and should be accepted. Please review their comments and update the paper accordingly before the conference. Thank you and good job!